# Performance and patients' satisfaction with the A7+TouchCare insulin patch pump system: A randomized controlled non-inferiority study

**Coralie Amadou**[1,2]*, **Vincent Melki**[3], **Jennifer Allain**[4], **Sylvaine Clavel**[5], **Didier Gouet**[6], **Lucy Chaillous**[7], **Bogdan Catargi**[8], **Pauline Schaeplynck-Belicard**[9], **Catherine Petit**[1,2], **Charles Thivolet**[10], **Alfred Penfornis**[1,2]

1 Service d'Endocrinologie, Diabétologie et Maladies Métaboliques, Centre Hospitalier Sud-Francilien de Corbeil-Essonnes, Corbeil-Essonnes, France, 2 Université Paris-Saclay, Gif-sur-Yvette, France, 3 Service de Diabétologie–Maladies Métaboliques—Nutrition, CHU Hôpital Rangueil, Toulouse, France, 4 Service d'Endocrinologie-Diabétologie, Centre Hospitalier de Gonesse, Gonesse, France, 5 Hôpital du Creusot, Le Creusot, France, 6 Hôpitaux La Rochelle, Ré, Aunis, La Rochelle, France, 7 Hôpital Nord Laennec, CHU Nantes, Saint-Herblain, France, 8 Hôpital Saint-André, CHU Bordeaux, Bordeaux, France, 9 Hôpital La conception, APHM, Marseille, France, 10 Centre du diabète Diab-e-Care—Hospices Civils de Lyon, Lyon, France

* coralie.amadou@u-psud.fr

## Abstract

### Background

We assessed the performance and patient satisfaction of a new insulin patch pump, the A7 +TouchCare (Medtrum), compared with the Omnipod system.

### Methods

This multicenter, randomized, open-label, controlled study enrolled 100 adult patients with type 1 or type 2 diabetes mellitus (A1C $\geq$ 6.5% and $\leq$ 9.5%, i.e., 48 to 80 mmol/mol) who were assigned with the Omnipod or with the A7+TouchCare pump for 3 months. The primary study outcome was the glucose management indicator (GMI) calculated with continuous glucose monitoring (CGM).

### Results

Premature withdrawals occurs respectively in 2 and 9 participants in the Omnipod and TouchCare groups. In the Per Protocol analysis, the difference in GMI between groups was 0.002% (95% confidence interval -0.251; 0.255). The non-inferiority was demonstrated since the difference between treatments did not overlap the pre-defined non-inferiority margin (0.4%). There was no significant difference in CGM parameters between groups. On average, patients in both groups were satisfied/very satisfied with the insulin pump system. Patients preferred Omnipod as an insulin management system and especially the patch delivery system but preferred the A7+TouchCare personal diabetes manager to control the system.

**Data Availability Statement:** Ethical and legal restrictions apply to the data as public data-sharing was not planned in the information letter given to

and signed by participants at inclusion in the study. Data-sharing would therefore require an amendment of the information given to patients after acceptance by the ethics committee and then participants.Here is the contact access for the ethics committee that approved the study: Comité de protection des personnes VI CHU G. MONTPIED Administration centrale 58 rue Montalembert BP 69 – 63003 CLERMONT FERRAND cedex 1, FRANCE Tél: +33 4 73 75 10 73 – Fax: +33 4 73 75 10 69 mail: cpp-sudest6@chu-clermontferrand.fr

**Funding:** Medtrum Technologies Inc. Shanghai supported this study. The funder had no role in the study design, data collection, and analysis. AP (principal investigator) takes full responsibility for the work, including the decision to submit and publish the manuscript. The findings and conclusions in this study are those of the authors and do not necessarily represent the sponsor's views.

**Competing interests:** C.A. declares congress invitations from Eli Lilly, Astrazeneca, and Sanofi; has received speaker honoraria from Diabeloop SA and Eli Lilly; has served on advisory board panels/ consulting services for Eli Lilly and Medtrum. V.M. has received congress invitations from: Novo-Nordisk, Isis, Sadir, Vitalaire, Orkyn, Medtronic, and speaker fees or consultancy for expertise from: Medtrum, ISIS, Novo-Nordis. J.A. has received congress invitations, speaker honoraria, and consultancy fees from Lilly, Sanofi and Medtronic and has served on advisory board panels for Novonordisk, Ypsomed, Medtrum and Glooko. S.C. has received no congress invitations, speaker honoraria, and consultancy fees over the last 24 months. D.G. has received congress invitations, speaker honoraria, and consultancy fees from Eli-Lilly, Novo-Nordisk, Sanofi-Diabète, Abbott, Medtronic, Astra Zeneca. L.C. declares that she has no engagement, fees or honoraria conflicting with her participation to the project. B.C. has received congress invitations, speaker honoraria, and consultancy fees from Medtronic Dexcom and Abbott. P.S.-B. has received conference invitations, honoraria and consultancies from Abbott Diabetes Care, Eli-Lilly, Novo-Nordisk, Roche Diabetes Care and Ypsomed. C.P. has received no congress invitations, speaker honoraria, and consultancy fees over the last 24 months. C.T. has received honoraria for consultancy, speaker for conferences and congress invitations, from Abbott, Janssen, Lilly, Medtronic, Roche Diabetes Care, Sanofi. A.P. has received congress invitations, speaker honoraria, and consultancy fees from Abbott, Diabeloop, Eli Lilly and Company, Insulet, Lifescan, Medtronic, Novo Nordisk, and Sanofi, and has

## Conclusions

This study showed that the A7+TouchCare insulin pump was as efficient as the Omnipod pump in terms of performance and satisfaction.

## Clinical trail registration

The study was registered in the ClinicalTrials.gov protocol register (NCT04223973).

## Introduction

Treatment of patients with type 1 (T1D) and some type 2 (T2D) diabetes mellitus is based on insulin therapy in order to mimic the physiological pancreatic secretion according to a basal/ bolus pattern obtained either by multiple daily injections (MDI) or continuous subcutaneous insulin infusion (CSII) with an external insulin pump. In a very large population-based study [1], the percentage of patients with T1D using insulin pump therapy increased from 1% in 1995 to 53% in 2017. There are many reasons for this rise, but the main ones are probably the effectiveness on glycemic control, safety, and the improvement of patients' quality of life [2, 3].

However, the indications for an external pump therapy for non-pregnant adults with T1D has been summarized by the French Society for the Study of Diabetes [4] as follows 1) A1C persistently > 7.5% (58 mmol/mol) despite intensified MDI (at least 3 injections per day) 2) Recurrent severe (more than one a year) or moderate (more than four a year) hypoglycemia, 3) Marked glycemic variability, 4) Variability in insulin needs, 5) Good metabolic control under MDI, but undermining the patient's social/professional life. For patients with T2D, the situations concerned are mostly failure of intensified MDI regimen and patients with insulin resistance or very high insulin requirements. Absolute contra-indications are infrequent and include severe psychiatric disorders, ischemic or proliferative retinopathy progressing rapidly (before laser treatment) and repeated scheduled exposition to high magnetic field.

During the current decade, there has been several innovations in the management of diabetes, including insulin pumps improvement. Pumps have become smaller, less invasive and easier to use. They now offer the possibility to be equipped with sensors and integrated algorithms allowing closed-loop insulin delivery systems ("artificial pancreas") [5].

Conventional insulin pumps deliver insulin continuously using tubing and external catheter. These pumps have proved to be effective as intensive therapy of diabetes by improving the glycemic control and reducing hypoglycemic events [3, 6]. By comparison, patch pumps are external pumps built without tubing and do not require the installation of a catheter [7, 8]. They also deliver insulin continuously, but the insulin delivery component of the system is a consumable managed by a personal diabetes manager (PDM). The insulin delivery reservoirs adhere to the skin with an adhesive patch that lasts up to 3 days. They are changed regularly, and these devices do not require any long-term maintenance. The absence of long tubing reduces the occurrence of adverse event associated with it, like insulin delivery modified by changing position [9]. Regarding occlusion events, the alarm is more immediate in new patch pumps design due to the absence of a catheter, and therefore allow earlier occlusion detection [10, 11]. Moreover, these systems are lighter, allowing less to carry on a day-to-day basis. Wearing such devices could lead to improved compliance and patients tend to prefer this kind of models [12–14]. However, they remain a minority in the supply, especially in France where the Omnipod® pump was until recently the only patch pump available.

served on advisory board panels for Abbott, Insulet, Medtrum, Novo Nordisk, and Sanofi. This does not alter our adherence to PLOS ONE policies on sharing data and materials.

The A7+TouchCare® insulin management system (Medtrum Technologies Inc. Shanghai) is a CE marked device (Class IIB) recently available in France and other countries. This patch pump is a small device designed for continuous subcutaneous delivery of insulin, made of a durable part of the pump that functions with a specific consumable. Furthermore, the A7+-TouchCare pump comes with a cloud-based data management system that can be operated from app and laptop providing real-time information.

The aim of this real-life study was to collect performance, safety, tolerability and patient' satisfaction with the A7+TouchCare pump system in patients with T1D and T2D after 3 months of use, in comparison with Omnipod® (Insulet Corp., Bedford, USA), a widespread used insulin patch pump system [7, 15].

## Materials and methods

### Study design and patients

This was a multicenter, randomized, open-label comparative study. The study was conducted between February 2020 and February 2021 in 8 hospital centers in France that recruited patients. The randomization was done in two parallel groups (1:1), with a stratified randomization by center with randomized blocks of 4. Randomization lists were produced by the CRO statistician with SAS Version 9.4. The randomization list was implemented in the Ennov Clinical eCRF.

To be enrolled in the study, patients had to fulfil the following inclusion criteria: type 1 or 2 diabetes mellitus, at least 18 years old, already using an Omnipod® insulin patch pump and the Abbott FreeStyle Libre® sensor version 1 (Abbott Diabetes Care, Alameda, CA), and with laboratory A1C $\geq 6.5\%$ and $\leq 9.5\%$ ($\geq 48$ mmol/mol and $\leq 80$ mmol/mol). The inclusion of patients already using a pump was done to facilitate recruitment into the study. Any type of rapid insulin could be used except faster aspart insulin (in line with the device CE marking at the time of the study) which could be substituted by any other rapid insulin (60 international units (IU) max per day). After screening, patients were randomly assigned to the investigational device (A7+TouchCare) or the Omnipod system and followed for 3 months.

### Pump characteristics

The A7+TouchCare pump dimensions are 56.3x33.3x13.3 mm and it is made of a durable part of the pump (the Pump Base) that functions with a specific consumable (Reservoir-Patch). The pump is strapped to the body through its adhesive base. The insulin delivery is made thanks to a 5-mm cannula in stainless steel with a 90˚ penetration into the dermis. The reservoir patches contain up to 200 units of rapid-acting insulin allowing delivery for up to 3 days. The pump is monitored and controlled via a wireless radiofrequency small PDM (76.2x48.4x9.4 mm) featured by its color touchscreen. The A7+TouchCare pump also offers the possibility of personalization of insulin rate and to reduce the basal rate to zero unit/hour (for pediatrics use). All patients in the A7+touchCare group received the relevant documentation and were trained by investigator for using the A7+TouchCare device, just like in real life.

### Continuous glucose monitoring

All patients were equipped with the Freestyle Libre sensor, a flash continuous glucose monitoring (CGM) system. This CGM requires scanning the sensor periodically for a continuous statement of glucose level. To obtain a complete visualization of glucose level on the last three months, the sensor was replaced every 14 days and scanned by the patient at least once every 8 hours.

## Clinical data collections

Clinical data were recorded by the investigators on an electronic case report form (eCRF) at Visit 1 (inclusion visit), Visit 2 (1 month) and Visit 3 (3 months). Data generated by the insulin pump systems and the CGM were recorded throughout the study and extracted from web platforms available for the study. CGM wearing rate, glucose metrics, and glycemic events were obtained automatically on the last 4 weeks at Visit 1 and Visit 2, and on the last 8 weeks at Visit 3. Time spent in range (70–180 mg/dL) (TIR), below range (< 70 mg/dL) (TBR) and above range (> 180 mg/dL) (TAR) were defined according to the recommendations from the international consensus on time in range [16].

Hypoglycemic events were defined as a glycemia under 70 mg/dL for at least 15 minutes, according to the Freestyle Libre sensor definition. Severe hypoglycemia was defined by severe cognitive impairment requiring external assistance for recovery.

Venous blood sample was taken at inclusion and at Visit 3 in order to determine the laboratory A1C.

The doses of insulin (basal and bolus) determined by the patient and injected by the pump were recorded by the medical device. These data were displayed and downloaded by the investigators using the online application provided by the manufacturer.

## Study outcomes

The primary study outcome was the glucose management indicator (GMI), an estimation of the laboratory A1C based on glucose parameters collected from CGM, on the last 10 weeks [17]. Other glucose metrics calculated automatically by the CGM software and recorded by investigators were secondary study outcomes as well as A1C measured by the laboratory.

An auto-questionnaire developed for the study was filled out by the patient at each visit to determine their satisfaction when using the insulin pump device. Existing validated questionnaires did not a priori allow sufficient exploration of the differences between the two pump models. The questionnaire was composed of 16 items (one question per item, except for item 10 that included 5 questions) concerning overall satisfaction and feelings about the blood glucose control (4 items); on the start-up and placement of the system (3 items); on the daily use of the system (3 items); on the management of the system with the PDM (4 items); and on the withdrawal/replacement of the system (2 items). For each question, the satisfaction scale went from 1 (very unsatisfied) to 5 (very satisfied), except for question 7 about pain for which 5 was painful and 1 was not painful at all.

At Visit 3, patients randomized in the A7+TouchCare group were also asked to indicate their preference for the insulin management system, insulin patch and the PDM on a scale from 1 to 5 where 1 indicated the preference for the A7+TouchCare pump and 5 the preference for Omnipod pump.

Local skin tolerability at the site of the pump (irritation/itching and redness) and other adverse events related to the study devices were recorded at each visit.

## Sample size

The sample size was calculated based on an average GMI level obtained using the Omnipod pump of 7.8% (62 mmol/mol) according to the scientific committee. The non-inferiority between the study insulin pump devices was defined using the commonly accepted non-inferiority margin of 0.4% in similar study related to diabetes treatments [18]. With these assumptions, a type-1 error of 2.5%, a power of 80% and a standard deviation fixed at 0.55, the calculated number of subjects necessary to demonstrate non-inferiority was 60 patients (30

patients in each group). Accounting for 20% of non-analyzable patients (e.g. due to major protocol deviations), the number of patients to be randomized was to be at least 75.

## Statistical analysis

Statistical analyses were computed with SAS Version 9.4 (SAS Institute Inc., Cary, USA). Four analysis populations were defined for this study: the safety population which was composed of all patients having used the investigational device; the intent-to-treat (ITT) population composed of all included and randomized patients having used the investigational device; the modified ITT (mITT) population composed of all ITT patients excluding patients who ended the study because of the COVID-19; the per protocol (PP) population composed of all ITT patients with no major protocol deviation. Continuous and categorical variables were summarized using usual statistics. Non-inferiority analysis was made by an analysis of covariance in the PP population with the medical device as an explanatory variable and adjusted on the baseline A1C estimation. Then non-inferiority was demonstrated if the difference (LSmeans) between treatments did not overlap the non-inferiority margin (0.4%). Superiority was tested and demonstrated in the presence of a difference of 0.3% in favor of the new pump. A sensitivity analysis was performed on mITT population with imputation using the median values in case of missing data. In addition, univariate analyses were performed on age in class (according to the median value), gender, diabetes mellitus type, center, and duration of diabetes. Variables with a p-value less than 0.05 were included in the multivariate model. All other comparisons between groups were performed using Student's t-test, a chi-squared test or a Fisher exact test.

The study was approved by an independent ethic committee and was registered in the French and European database of clinical studies (ID-RCB: 2019-A02566-51) and in the ClinicalTrials.gov protocol register under NCT04223973 (study start date: January 29, 2020; primary completion date: March 1, 2021; study completion date: June 1, 2021).

The study was performed in compliance with the General Data Protection Regulation (GDPR) and with the Data Privacy Committee CNIL MR001. The study was carried out in accordance with the Declaration of Helsinki. All patients were carefully informed about the study and gave written informed consent to participate.

## Results

### Patients disposition and baseline characteristics

Overall, 100 patients were included and 93 were randomized and used the allocated study device (ITT Population): 45 patients in the Omnipod group and 48 patients in the A7+TouchCare group. Among them, 78 completed the study: 43 (95.6%) in the Omnipod group and 35 (72.9%) in the A7+Touchcare group. The main reasons for premature study withdrawal were the Covid-19 context (4 patients ended the study because of the COVID-19 context and one withdrew consent), intolerance (4 patients), device dysfunction (3 patients), consent withdrawal (1 patient), or another reason (2 patients). The PP population was therefore composed of 78 patients (43 patients in the Omnipod group and 35 patients in the A7+TouchCare group). The mITT population was composed of 89 patients (45 in the Omnipod group and 44 in the A7+TouchCare group). (Fig 1)

As shown in Table 1, there was no significant or clinically relevant difference between groups in baseline characteristics including age, gender ratio, body mass index (BMI), time since diabetes diagnosis, time since the initiation of insulin pump treatment, and insulin daily doses. The last laboratory A1C value (± standard deviation, SD) was 7.8±0.8% (62 mmol/mol) in the Omnipod group and 7.8±0.7% in the A7+TouchCare group. No centre effect was found.

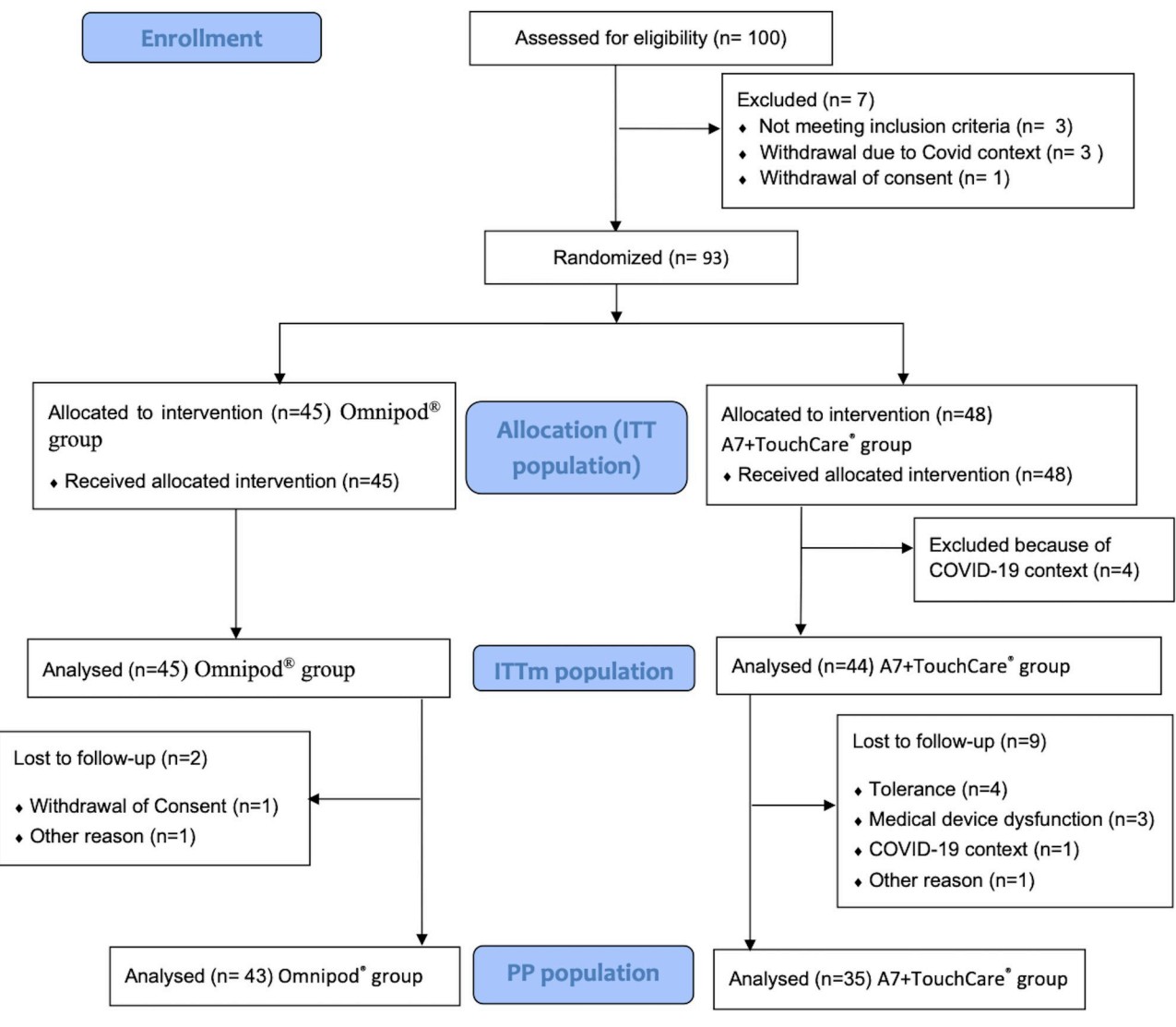

**Fig 1. Flow chart.**

### Primary outcome

In the PP population, the GMI after 3 months of use (Visit 3) was similar in the Omnipod group (mean ± SD: 7.4±0.8%, 57 mmol/mol) and the A7+TouchCare group (7.3±0.8%, 56 mmol/mol). As summarized in Table 2, the difference in GMI between device groups (Omnipod–A7+TouchCare) was 0.002% with a 95% CI of [-0.251; 0.255] in the PP population. The non-inferiority was demonstrated (p < 0.001) since the 95% CI of the difference did not overlap the non-inferiority margin (0.4%). This was confirmed in the mITT population. Age was the only significant covariate associated with the GMI (p = 0.0265) contrary to gender, type of diabetes, centre, and duration of diabetes (p > 0.10). Non-inferiority was also demonstrated in the PP Population in age-adjusted non-inferiority analysis. In all analyses, superiority was not demonstrated.

**Table 1. Characteristics of study participants at baseline (ITT population).**

| | | Omnipod N = 45 | A7+Touchcare N = 48 |
|---|---|---|---|
| **Age (years)** | Mean ± SD | 50.3 ± 13.6 | 49.4 ± 11.7 |
| | Min.; Max. | 24; 75 | 22; 72 |
| **Gender** | Female | 29 (64.4%) | 24 (50.0%) |
| | Male | 16 (35.6%) | 24 (50.0%) |
| **BMI (kg/m$^2$)** | Mean ± SD | 25.4 ± 4.3 | 26.5 ± 4.1 |
| | Min.; Max. | 17.9; 37.3 | 17.6; 35.1 |
| **Diabetes type** | Type 1 | 43 (95.6%) | 41 (85.4%) |
| | Type 2 | 2 (4.4%) | 7 (14.6%) |
| **Time since diabetes diagnosis (yrs)** | Mean ± SD | 22.2 ± 11.5 | 22.7 ± 11.1 |
| | Min.; Max. | 3.7; 47.7 | 2.7; 44.0 |
| **Time since initiation of** | Mean ± SD | 7.1 ± 6.7 | 5.4 ± 4.3 |
| **insulin pump treatment (yrs)** | Min.; Max. | 0.1; 35.4 | 0.8; 19.0 |
| **Laboratory A1C (%)*** | Mean ± SD | 7.8 ± 0.8 | 7.8 ± 0.7 |
| | Min.; Max. | 7; 9 | 7; 10 |
| **Basal insulin dose (IU/day)** | Mean ± SD | 16.1 ±5.9 | 18.7 ± 10.0 |
| | Min.; Max. | 6; 34 | 2; 43 |
| **Bolus insulin dose (IU/day)** | Mean ± SD | 21.5 ± 9.0 | 18.7 ± 10.9 |
| | Min.; Max. | 6; 41 | 1; 51 |

Data are n (%) or means±SD; ND: not determined;

* Last laboratory A1C value of less than 1 month; No significant difference was observed between the two groups.

## Laboratory A1C

In the mITT population, there was no relevant variation in laboratory A1C which remained stable between inclusion (mean ± SD: 7.8±0.7%, 62 mmol/mol) and Visit 3 (7.7±0.7%, 61 mmol/mol) without statistically significant difference (p = 0.565) between the two groups.

**Table 2. Glucose management indicator (GMI): Non inferiority analysis at Visit 3 (Month 3).**

| | Omnipod | A7+Touchcare |
|---|---|---|
| **Per protocol analysis** | N = 43 | N = 35 |
| Estimated mean* at Month 3 | 7.347 (0.085) | 7.345 (0.094) |
| Estimated between-group difference [95%CI] | 0.002 [-0.251; 0.255] | |
| p-value for non-inferiority | <0.001 | |
| **mITT analysis** | N = 45 | N = 44 |
| Estimated mean* at Month 3 | 7.388 (0.092) | 7.287 (0.093) |
| Estimated between-group difference [95%CI] | 0.101 [-0.158; 0.360] | |
| P-value for non-inferiority | <0.001 | |
| **Per protocol analysis adjusted for age** | N = 43 | N = 35 |
| Estimated mean* at Month 3 | 7.344 (0.086) | 7.341 (0.095) |
| Estimated between-group difference [95%CI] | 0.003 [-0.251; 0.257] | |
| P-value for non-inferiority | <0.001 | |

* LSmeans (SEM); Difference Omnipod—A7+Touch Care; Missing values in the mITT population were replaced by the median value.

## Freestyle Libre data

The mean (± SD) CGM wearing rate was not significantly different between the Omnipod group and the A7+TouchCare group at each visit: 91.3±11.4% vs. 90.0±13.6% (p = 0.625) at inclusion, 94.5±7.0% vs.91.6±11.8% (p = 0.200) at Visit 2, and 91.3±11.3% vs. 89.2±19.4% (p = 0.568) at Visit 3.

Regardless of devices, the average glucose level remained stable between inclusion (165.2 ±27.2 mg/dL) and Visit 3 (164.2±22.7 mg/dL) without statistically significant difference between groups (p = 0.932) (Table 3). The coefficient of variation of glucose value was 40.6% at Visit 3 similar to baseline (40.8%) without statistically significant difference between groups (p = 0.649).

The mean number of hypoglycemia events per week, regardless of device, remained stable between inclusion (5.6±3.3) and Visit 3 (5.5±3.6) without difference between groups (p = 0.602, at Visit 3) (Table 3). The TBR, TIR and TAR were 6.3%, 57.5% and 36.2% at Visit 3 quite similar to values at inclusion (6.4%, 56.0%, and 37.6%, respectively). There was no statistically significant difference between device groups in hypoglycemic events, or percentage of time spent in TBR, TAR and TAR at each visit.

## Insulin dose administered

As shown in Table 4, there was no statistically significant difference between groups in basal, bolus or total daily dose of insulin administered. At Visit 3, the number of bolus per day was 4.1±1.3 in the Omnipod group and 4.2±1.7 in the A7+TouchCare group (p = 0.758).

## Patients' satisfaction and preference

Results of the satisfaction questionnaire at Visit 2 and Visit 3 in the mITT population are provided in Table 5. Overall, the satisfaction for the A7+TouchCare was quite similar to Omnipod. At visit 3, the mean satisfaction score was ≥ 4 (satisfied/very satisfied) for 16/19 questions in the Omnipod group and 14/19 questions in the A7+TouchCare group. On average, patients were more satisfied about pump piloting with the A7+TouchCare pump PDM (questions 11, 12, 13, & 14). This was based on PDM ease of use as understanding of the instruction displayed (questions 12 and 14) and in terms of discretion for the use of the system in public (question 13). The A7+TouchCare pump PDM was gaining better score than the Omnipod pump PDM regarding the patient satisfaction whatever the visit.

At Visit 3, patients randomized in the A7+TouchCare group had to indicate their preference for the insulin pump device on a scale from 1 to 5 where 1 indicated the preference for the A7+TouchCare pump and 5 the preference for Omnipod pump. The preference score was 3.2±1.4 for the insulin management system, 3.7±1.5 for the patch insulin delivery system, and 1.9±1.3 for the PDM.

## Device-related adverse events

Irritation/itching were reported in 26.7% and 27.1% of patients in the Omnipod and the A7 +Touch Care, respectively. Redness were reported in 24.4% and 37.5% of patients in the Omnipod and the A7+Touch Care, respectively. As shown in S1 Table, there was no statistically significant differences between device groups (p = 0.964 and p = 0.174, respectively). Other related adverse events were more frequently (p = 0.010) reported in the A7+TouchCare group (31.3%) compared with the Omnipod group (8.9%). This was mainly due to more frequent bleeding or pain at the injection site, error using the device, or connection problems in the A7+TouchCare group.

**Table 3. Continuous glycaemic parameters at each visit (ITTm population).**

| | | Omnipod | A7+TouchCare | p-value |
|---|---|---|---|---|
| **Average glucose level (mg/dL)** | V1 | N = 45 | N = 44 | 0.784 |
| | | 164.4 ± 22.7 | 166.0 ± 31.4 | |
| | V2 | N = 43 | N = 37 | 0.560 |
| | | 163.3 ± 22.3 | 160.4 ± 21.9 | |
| | V3 | N = 43 | N = 36 | 0.932 |
| | | 164.4 ± 21.8 | 164.0 ± 24.0 | |
| **Standard deviation of glucose values (mg/dL)** | V1 | N = 45 | N = 44 | 0.970 |
| | | 67.2 ± 15.3 | 67.4 ± 16.4 | |
| | V2 | N = 41 | N = 36 | 0.912 |
| | | 66.1 ± 16.0 | 66.5 ± 15.2 | |
| | V3 | N = 42 | N = 35 | 0.895 |
| | | 66.6 ± 16.6 | 67.1 ± 14.7 | |
| **Coefficient of variation (%) of glucose values** | V1 | N = 45 | N = 44 | 0.564 |
| | | 41.3 ± 7.5 | 40.4 ± 7.6 | |
| | V2 | N = 42 | N = 35 | 0.546 |
| | | 40.7 ± 6.9 | 41.7 ± 7.7 | |
| | V3 | N = 42 | N = 35 | 0.649 |
| | | 40.2 ± 7.0 | 41.0 ± 8.0 | |
| **Hypoglycaemic events (per week)** | V1 | N = 45 | N = 44 | 0.816 |
| | | 5.7 ± 3.3 | 5.6 ± 3.4 | |
| | V2 | N = 43 | N = 37 | 0.383 |
| | | 5.5 ± 3.1 | 6.2 ± 4.1 | |
| | V3 | N = 43 | N = 36 | 0.602 |
| | | 5.3 ± 3.2 | 5.8 ± 4.1 | |
| **Time below range (TBR) (%)** | V1 | N = 45 | N = 44 | 0.271 |
| | | 7.0 ± 5.0 | 5.9 ± 4.2 | |
| | V2 | N = 43 | N = 37 | 0.973 |
| | | 7.2 ± 8.0 | 7.1 ± 5.5 | |
| | V3 | N = 43 | N = 36 | 0.223 |
| | | 5.6 ± 3.9 | 7.1 ± 6.4 | |
| **Time in range (TIR) (%)** | V1 | N = 45 | N = 44 | 0.685 |
| | | 56.6 ± 12.9 | 55.4 ± 14.7 | |
| | V2 | N = 43 | N = 37 | 0.760 |
| | | 58.1 ± 12.6 | 58.9 ± 12.1 | |
| | V3 | N = 43 | N = 36 | 0.512 |
| | | 58.4 ± 13.1 | 56.5 ± 12.3 | |
| **Time above range (TAR) (%)** | V1 | N = 45 | N = 44 | 0.500 |
| | | 36.6 ± 13.8 | 38.7 ± 15.8 | |
| | V2 | N = 43 | N = 37 | 0.788 |
| | | 34.7 ± 13.9 | 33.9 ± 12.4 | |
| | V3 | N = 43 | N = 36 | 0.891 |
| | | 36.0 ± 13.4 | 36.4 ± 13.4 | |

Data are means ± SD

One serious adverse event related to the A7+ Touch Care pump (3 episodes of ketonemia) was reported during the study. The patient was not life-threatened and the resolution of this

**Table 4. Insulin doses administered at each visit (mITT population).**

| | | Omnipod | A7+TouchCare | p-value |
|---|---|---|---|---|
| **Total insulin dose (basal + bolus) (IU/day)** | V1 | N = 45<br>37.6 ± 11.8 | N = 44<br>36.8 ± 15.0 | 0.788 |
| | V2 | N = 41<br>38.0 ± 11.5 | N = 37<br>38.9 ± 14.1 | 0.757 |
| | V3 | N = 42<br>37.3 ± 12.6 | N = 36<br>37.3 ± 14.9 | 0.984 |
| **Basal insulin dose (IU/day)** | V1 | N = 45<br>16.1 ± 5.9 | N = 44<br>18.6 ± 9.6 | 0.153 |
| | V2 | N = 41<br>16.4 ± 5.6 | N = 37<br>19.7 ± 9.6 | 0.073 |
| | V3 | N = 42<br>16.3 ± 5.8 | N = 36<br>19.4 ± 9.3 | 0.088 |
| **Bolus insulin dose (IU/day)** | V1 | N = 45<br>21.5 ± 9.0 | N = 44<br>18.2 ± 11.0 | 0.135 |
| | V2 | N = 41<br>21.5 ± 9.2 | N = 37<br>19.1 ± 9.2 | 0.257 |
| | V3 | N = 42<br>21.1 ± 9.8 | N = 36<br>17.9 ± 9.8 | 0.160 |
| **Number of bolus per day** | V1 | N = 45<br>4.1 ± 1.2 | N = 44<br>3.8 ± 1.6 | 0.227 |
| | V2 | N = 40<br>4.2 ± 1.3 | N = 37<br>4.3 ± 1.8 | 0.800 |
| | V3 | N = 42<br>4.1 ± 1.3 | N = 36<br>4.2 ± 1.7 | 0.758 |

Data are means±SD

event was rapid and positive. However, the patient was withdrawn from the study and resumed the use of his previous pump.

## Discussion

The A7+TouchCare is an insulin patch pump system recently available on the market in some European countries. The aim of the study was to determine the clinical performance and patients' satisfaction of the A7+TouchCare compared to the Omnipod insulin patch pump, which was the only patch pump available in France to date. Here, we demonstrated similar performance to achieve glycemic control with the A7+TouchCare.

This is the first report of clinical data on this new device, through a randomized study that included middle aged adult patients (mean age: 50 years-old), mostly with T1D and a long duration of diabetes (> 20 years). Glycemic control was suboptimal at baseline (mean A1c: 7.8%, 62 mmol/mol), considering that participants were already users of an insulin pump and a flash CGM device. Moreover, it is interesting to note a mean BMI of 25.4 and 26.5 kg/m2 for Omnipod and A7+Touchcare group, respectively, despite a total daily dose of insulin limited to 60 units per day. Therefore, these baseline characteristics are representative of a real-world population [19].

The Omnipod patch pump is known to provide improved glycemic control and preference compared to MDI or conventional insulin pumps in patients with diabetes [12–14]. Although various patch pumps systems have been recently developed [15], there is a lack of head-to-head comparison between different insulin patch pump systems. Here, we demonstrated that the performance of the A7+TouchCare was non inferior compared with the Omnipod patch

**Table 5. Satisfaction questionnaire at Visit 2 and Visit 3 (mITT population).**

| | Visit 2 | | Visit 3 | |
|---|---|---|---|---|
| | Omnipod | A7+ TouchCare | Omnipod | A7+ TouchCare |
| **1. How satisfied are you with the control of your blood sugar obtained with the pump?** | N = 42 | N = 35 | N = 41 | N = 34 |
| | 4.0 ± 0.9 | 4.0 ± 0.9 | 4.2 ± 0.7 | 4.0 ± 0.9 |
| **2. How would you qualify the management of your treatment with the pump?** | N = 43 | N = 36 | N = 43 | N = 36 |
| | 4.3 ± 0.7 | 4.0 ± 1.0 | 4.2 ± 0.7 | 3.8 ± 1.0 |
| **3. Overall, how satisfied were you with the pump?** | N = 42 | N = 36 | N = 43 | N = 36 |
| | 4.1 ± 0.9 | 3.7 ± 0.9 | 4.1 ± 0.7 | 3.5 ± 1.2 |
| **4. How would you rate your level of daily comfort after wearing the pump?** | N = 43 | N = 36 | N = 43 | N = 36 |
| | 4.2 ± 0.9 | 3.7 ± 1.0 | 3.9 ± 0.9 | 3.4 ± 1.1 |
| **5. Was it easy for you to insert the needle of the syringe into the reservoir of the pump?** | N = 43 | N = 35 | N = 43 | N = 36 |
| | 4.6 ± 0.7 | 4.3 ± 0.9 | 4.6 ± 0.7 | 3.9 ± 1.2 |
| **6. Was it easy to put down the pump you just used?** | N = 43 | N = 35 | N = 42 | N = 36 |
| | 4.5 ± 0.7 | 4.2 ± 0.8 | 4.5 ± 0.7 | 4.0 ± 0.9 |
| **7. Was it painful to insert the pump needle that you just used in the past few weeks?** | N = 43 | N = 34 | N = 43 | N = 36 |
| | 2.2 ± 1.2 | 2.4 ± 1.4 | 2.6 ± 1.3 | 2.6 ± 1.4 |
| **8. In general, would you say giving insulin with the pump is convenient?** | N = 42 | N = 34 | N = 42 | N = 36 |
| | 4.5 ± 0.8 | 4.3 ± 0.9 | 4.5 ± 0.6 | 4.2 ± 0.8 |
| **9. Would you say that delivering boluses with the pump is easy?** | N = 43 | N = 34 | N = 43 | N = 36 |
| | 4.7 ± 0.5 | 4.4 ± 0.7 | 4.6 ± 0.5 | 4.4 ± 0.8 |
| **10a. Programming basal rates: Would you say this pump feature is easy to use?** | N = 33 | N = 24 | N = 33 | N = 25 |
| | 4.2 ± 0.7 | 4.1 ± 0.7 | 4.2 ± 0.8 | 4.3 ± 0.7 |
| **10b. Programming bolus: Would you say this pump feature is easy to use?** | N = 32 | N = 28 | N = 31 | N = 26 |
| | 4.5 ± 0.7 | 4.5 ± 0.6 | 4.4 ± 0.8 | 4.6 ± 0.6 |
| **10c. Temporary basal rate stop: Would you say this pump feature is easy to use?** | N = 35 | N = 25 | N = 36 | N = 30 |
| | 4.5 ± 0.7 | 4.5 ± 0.6 | 4.5 ± 0.7 | 4.4 ± 1.0 |
| **10d. Bolus calculator/bolus assistant: Would you say this pump feature is easy to use?** | N = 26 | N = 24 | N = 28 | N = 25 |
| | 4.1 ± 1.0 | 4.6 ± 0.6 | 4.2 ± 0.9 | 4.5 ± 0.7 |
| **10e. Insulin Active (IOB): Would you say this pump feature is easy to understand and to use?** | N = 18 | N = 22 | N = 22 | N = 25 |
| | 4.2 ± 1.1 | 4.3 ± 0.8 | 3.9 ± 1.2 | 4.4 ± 0.7 |
| **11. How satisfied are you overall with using the pump PDM?** | N = 43 | N = 35 | N = 43 | N = 35 |
| | 3.9 ± 0.9 | 4.3 ± 0.6 | 4.1 ± 0.8 | 4.1 ± 0.9 |
| **12. How do you qualify the ease of use of the pump's PDM?** | N = 43 | N = 35 | N = 43 | N = 35 |
| | 4.1 ± 0.9 | 4.4 ± 0.7 | 4.2 ± 0.8 | 4.2 ± 1.0 |
| **13. How to qualify the discretion of use of the PDM of the pump?** | N = 42 | N = 34 | N = 43 | N = 36 |
| | 3.6 ± 1.2 | 4.3 ± 1.0 | 3.8 ± 1.0 | 4.4 ± 0.8 |
| **14. Was it easy to understand the instructions displayed on the pump PDM?** | N = 43 | N = 34 | N = 43 | N = 36 |
| | 4.3 ± 0.7 | 4.3 ± 0.7 | 4.3 ± 0.6 | 4.4 ± 0.8 |
| **15. Was it easy to remove the cannula from the pump?** | N = 43 | N = 34 | N = 43 | N = 36 |
| | 4.6 ± 0.7 | 3.9 ± 1.1 | 4.6 ± 0.7 | 3.9 ± 1.2 |
| **16. How easy has it been to change the pump pods you have been wearing for the past few weeks?** | N = 43 | N = 34 | N = 42 | N = 36 |
| | 4.6 ± 0.5 | 4.1 ± 0.9 | 4.7 ± 0.5 | 4.0 ± 1.0 |

Data are means±SD

For each question, the scale went from 1 (very unsatisfied) to 5 (very satisfied), except for Question 7 about pain for which 5 was painful and 1 was not painful at all.

system, based on the GMI. Other glycemic parameters based on CGM were also not significantly different between the two insulin patch pumps, and the total dose of insulin administered including the basal and bolus infusion were comparable with both devices.

A1C is an important indicator of long-term glycemic control with the ability to reflect the cumulative glycemic history of the preceding two to three months. For this study, we made the choice of A1C evaluation obtained from sensor measurements (GMI), which is considered as a reliable estimation of the average glucose value [16, 17]. In a secondary analysis, we confirmed that there was no statistically significant difference between insulin pumps in laboratory A1C after 3 months of use. However, A1C is not informative of the percentage of time spent in hyper or hypoglycemia and does not reflect the day-to-day glucose variability. Therefore, CGM provides crucial data as the percentage of time spent in target range (70–180 mg/dL) and time spent below range (<70 mg/dL). One major goal for effective and safe glucose control is to reach a percentage of time in range of at least 70% while reducing the time below range to less than 4% [16]. In the present study, in patients with T1D or T2D, the TBR after 3 months regardless of the insulin patch pump used was 6.3±5.2%, and the TIR was 57.5±12.7%, similar to baseline values (6.4±4.6% and 56.0±13.7%, respectively). These results were closed but suboptimal compared to the recommended target values (4% and 70%, respectively) but the observation period was only 3 months. There was no statistically significant difference between groups in these metrics. The study also showed that glycemic metrics including average glucose level and variability remained stable in the A7+TouchCare group and the Omnipod group without significant difference between groups. Glycemic variability is a therapeutic challenge and a frequent issue for patients with long lasting diabetes. In addition to causing glycemic excursions or severe hypoglycemia, it places a heavy mental burden on the patient [20]. Predictive low-glucose suspend systems and hybrid closed-loop systems are the ideal solution to address this issue with a real-time adjustment of insulin infusion. These systems represent the future of insulin pump use and are already spreading with a large amount of very positive results from "real-world" data [21–24]. In this context, Medtrum has already improved its TouchCare pump by developing a next generation pump. The A8 version, also known as Nano, is designed to address most of the drawbacks reported in the present study [25].

An important aspect of continuous insulin infusion device is to ensure that people interact properly with technology [26]. Thus, patient's treatment satisfaction and preference are important consideration. The satisfaction questionnaire in our study showed that patients in both groups were satisfied/very satisfied with the patch delivering insulin and the PDM. Although differences between groups in each question may be not relevant, higher level of satisfaction was obtained in the A7+TouchCare group for some questions related to the use of the PDM system, such as ease of use, discretion of use, or calculator/bolus assistant usage (advanced functionalities). On the contrary, lower level of satisfaction was obtained for the ease of usage related to the insulin reservoir patch, or related to cannula including comfort on daily wear, insertion or removal. The preference questionnaire showed that patients preferred the Omnipod POD as insulin delivery device although one advantage of the A7+TouchCare is that the insulin reservoir can be replaced without having the need to replace the electronic part (located in the pump base). On the other hand, participants preferred the A7+TouchCare PDM for piloting the pump and advanced functionalities setting. However, it should be noted that all patients had been using the Omnipod pump for several months or years before the study and were thus familiar with its use. Moreover, patients in the A7+TouchCare group received only minimal training at inclusion, without any renewal of this training as it is usually done in the French daily practice by certified diabetic nurses educator from home care companies that detail and take care of patients at their home on regular basis. Finally, the Covid 19 context did not help neither as some intermediate visit (such as visit 2) where led on remote mode.

During the study period, some device errors were reported. Dysfunction reported are consistent with feedback from patients using this type of device in real life [26]: occlusion and detachment were relatively frequent in both groups but these were generally minor defects without severe consequences in terms of safety (excepted one case of ketosis leading to study withdrawal). In line with a joint statement of the European association for the study of diabetes and the American diabetes association diabetes technology working group [26], there is, for the Medtrum pump (as like other devices), a need for education and training of patients in learning how properly fixing the pump base to the reservoir-patch and identifying the best place for patch setting and skin preparation to increase adherence of patch.

Patch pumps adheres to the skin with adhesives which can induce more or less harmless skin reactions compared to conventional insulin pumps, as they require a larger adhesive area. In this study, adverse events related to the study devices were relatively frequent in both patch pump groups, but as is the case in current practice and reported by patients in real-life use [26, 27]. This included itching/irritation and redness at injection site which were specifically investigated for this study. Itching/irritation and redness were relatively frequent but without statistically significant difference between groups. No allergic reaction or infection at the administration site was reported.

One major limitation of this study is that patients were experienced in using insulin patch pump. They were equipped with an insulin pump since several years and all patients were familiar with diabetes-related technology; thus, results should be generalized with caution to other novice diabetes populations. In order to facilitate the recruitment, we decided to include all patients who already used the Omnipod pump before randomization (and thus without a change of their pump). On the other hand, patients in the A7+TouchCare group had only minimal training in using the new insulin patch pump system. This may have an impact on the study outcomes including performance, notification of malfunctions or adverse effects, satisfaction, or preference. This also draws the attention on being more specific in patients training and for the train-the-trainer programmes [26]. Obviously, it was not possible to mask the study device, and this study was designed as an open-label study. This may have an impact on the recording of adverse events, especially mild cases. Finally, although patients with T2D were included in this study, their number was too small to generalize the study results in these patients.

## Conclusion

Both Omnipod and A7+TouchCare pumps showed similar performance to achieve glycemic control in patients with diabetes and the level of satisfaction was quite similar between groups. Although patients preferred the Omnipod as insulin management system and more specifically the patch delivery system (POD) of the Omnipod pump, they seem to prefer the PDM of the A7+TouchCare system for controlling and managing their pump. Overall, the A7+TouchCare may be a valuable alternative to Omnipod or other insulin patch pumps in some patients with diabetes.

## Supporting information

**S1 Table. Skin tolerability and adverse events between inclusion and Visit 3 (Safety population).**
(DOCX)

**S1 Checklist. CONSORT 2010 checklist of information to include when reporting a randomised trial\*.**
(DOC)

**S1 File.**
(PDF)

**S2 File.**
(PDF)

## Acknowledgments

The authors would like to thank CRO Axonal-Biostatem (Castries, France) for study management, data management and statistics, and Thierry Radeau Consulting (Epinay-Sous-Senart, France) for medical writing support in accordance with Good Publication Practice (GPP3) guidelines (https://www.ismpp.org/gpp3).

## Author Contributions

**Investigation:** Vincent Melki, Jennifer Allain, Sylvaine Clavel, Didier Gouet, Lucy Chaillous, Bogdan Catargi, Pauline Schaeplynck-Belicard, Catherine Petit, Charles Thivolet, Alfred Penfornis.

**Writing – original draft:** Coralie Amadou.

**Writing – review & editing:** Coralie Amadou, Vincent Melki, Jennifer Allain, Sylvaine Clavel, Didier Gouet, Lucy Chaillous, Bogdan Catargi, Pauline Schaeplynck-Belicard, Catherine Petit, Charles Thivolet, Alfred Penfornis.

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
