## [Decision Letter · Decision Letter 0]

4 Nov 2022

PONE-D-22-17489Performance and patients’satisfaction with the A7+TouchCare insulin patch pump system: a randomized controlled non-inferiority studyPLOS ONE

Dear Dr. Amadou,

Thank you for submitting your manuscript to PLOS ONE. After careful consideration, we feel that it has merit but does not fully meet PLOS ONE’s publication criteria as it currently stands. Therefore, we invite you to submit a revised version of the manuscript that addresses the points raised during the review process. Your manuscript has been assessed by two reviewers and their reports are appended below.  The reviewers comment that the discussion section should comment on device disfunction and other adverse events reported for the pump. In addition, the reviewers have requested further detail and/or clarifications regarding the statistical analyses reported in this study. Furthermore, the journal's editorial team has noted that the ethics statement in this article reports that "The study was approved by an independent ethic committee" only. Please provide more details regarding the ethics committee that approved this study.  Could you please revise the manuscript to carefully address the concerns raised?

We look forward to receiving your revised manuscript.

Kind regards,

Maria Elisabeth Johanna Zalm, Ph.D

Editorial Office

PLOS ONE

Journal Requirements:

YES. The study was funded by Medtrum France (Boulogne Billancourt, France).

I have read the journal's policy and the authors of this manuscript have the following competing interests: C.A. has received writing/consulting honoraria from Medtrum for the present study, declares congress invitations from Eli Lilly, Astrazeneca, and Sanofi; has received speaker honoraria from Diabeloop SA and Eli Lilly;  has served on advisory board panels for Eli Lilly. V.M. has received congress invitations from: Novo-Nordisk, Isis, Sadir, Vitalaire, Orkyn, Medtronic, and speaker fees or consultancy for expertise from: Medtrum, ISIS, Novo-Nordis. J.A. has received congress invitations, speaker honoraria, and consultancy fees from Lilly, Sanofi and Medtronic and has served on advisory board panels for Novonordisk, Ypsomed, Medtrum and Glooko. S.C.  has received no congress invitations, speaker honoraria, and consultancy fees over the last 24 months. D.G.  has received congress invitations, speaker honoraria, and consultancy fees from  Eli-Lilly, Novo-Nordisk, Sanofi-Diabète, Abbott, Medtronic, Astra Zeneca. L.C. declares that she has no engagement, fees or honoraria conflicting with her participation to the project. B.C. has received congress invitations, speaker honoraria, and consultancy fees from Medtronic Dexcom and Abbott. P.S.-B. has received conference invitations, honoraria and consultancies from Abbott Diabetes Care, Eli-Lilly, Novo-Nordisk, Roche Diabetes Care and Ypsomed. C.P. has received no congress invitations, speaker honoraria, and consultancy fees over the last 24 months. C.T. has received honoraria for consultancy, speaker for conferences and congress invitations, from Abbott, Janssen, Lilly, Medtronic, Roche Diabetes Care, Sanofi. A.P. has received congress invitations, speaker honoraria, and consultancy fees from Abbott, Eli Lilly and Company, Lifescan, Medtronic, Novo Nordisk, and Sanofi, and has served on advisory board panels for Abbott, Insulet, Medtrum, Novo Nordisk, and Sanofi.

Reviewers' comments:

Reviewer's Responses to Questions

**Comments to the Author**

1. Is the manuscript technically sound, and do the data support the conclusions?

Reviewer #1: Yes

Reviewer #2: Partly

2. Has the statistical analysis been performed appropriately and rigorously? 

Reviewer #1: Yes

Reviewer #2: Yes

3. Have the authors made all data underlying the findings in their manuscript fully available?

Reviewer #1: Yes

Reviewer #2: No

4. Is the manuscript presented in an intelligible fashion and written in standard English?

Reviewer #1: Yes

Reviewer #2: Yes

5. Review Comments to the Author

Reviewer #1: The article “Performance and patients’satisfaction with the A7+TouchCare insulin patch pump

system: a randomized controlled non-inferiority study” by Amadou and colleagues describes a clinical study with insulin pumps. The study is well-designed and results are presented in a comprehensive manner. Thank you for this nice contribution. I only have some minor comments:

General: I would recommend using the term “glucose management indicator (GMI)” instead of “estimated A1C” as this is state of the art and the methods section references the article by Bergenstal et al. that introduced the GMI

Methods: Obviously there was no inclusion criterion that describes a minimum required duration of pump use before study start. According to Table 1 the minimum duration was 0.1 years. This appears too short to be able to differentiate study outcomes from improvements in HbA1c due to change from MDI to CSII.

Results and Discussion: It seems like there is a kind of safety issue with the A7+TouchCare pumps, because 7 patients dropped out due to intolerance or device dysfunction and in general, there were more adverse events with this pump. I think this is should be paid more attention in the discussion of the article.

Author contributions: All investigators all listed as authors, but most of them did not make any further contributions to the manuscript. This is not conform with the ICMJE authorship criteria; e.g. critically revising the manuscript is another required contribution.

Reviewer #2: Here are some comments:

Statistical methods:

“In addition, univariate analyses were performed on age in class (according to

the median value), gender, diabetes mellitus type, center, and duration of diabetes.Variables with a p-value less than 0.05 were included in the multivariate model using a stepwise strategy.” My comment here is that conclusions on these variables should be drawn from a full model i.e. a multiple model including all variables. The stepwise strategy is not needed and if any adjustment for the primary analysis is done, why not adjust for all?

Table 1. Why include statistical tests. Since this is a randomized trial this is not needed. Read about “the table 1 fallacy” and point 15 in the consort checklist http://www.consort-statement.org/checklists/view/32--consort-2010/510-baseline-data.

How was the p-value for non-inferiority calculated (table 2)? Probably through the lsmeans statement, include this in the methods.

Table S1: Please indicate which test that is used and double check the p-values, is the result for “Redness” correct?

Why is the “other adverse events” not discussed and is the conclusion “Overall, the A7+TouchCare may be a safe and valuable alternative to Omnipod..” correct?

Minor: The following sentence is repeated: “The inclusion of patients already using a pump was done to facilitate recruitment into the study. The inclusion of patients already using a pump was done to facilitate recruitment into the study.”

6. PLOS authors have the option to publish the peer review history of their article (what does this mean?). If published, this will include your full peer review and any attached files.

Reviewer #1: No

Reviewer #2: No

---

## [Author Response · Author response to Decision Letter 0]

28 Feb 2023

Dear Editor,

The manuscript has been revised. 

You will find answers to each comment below (italics and bold).

Major and minor corrections (responding to comments) have been highlighted in yellow in the revised manuscript (“Revised Manuscript with Track Changes”). We also provide a final clean version (“'Manuscript”).

We are confident that the quality of the manuscript is now improved, and we hope our revised version will be considered suitable for publication. 

“The reviewers comment that the discussion section should comment on device dysfunction and other adverse events reported for the pump. In addition, the reviewers have requested further detail and/or clarifications regarding the statistical analyses reported in this study.” 

We made a detailed response for each comment.

Furthermore, the journal's editorial team has noted that the ethics statement in this article reports that "The study was approved by an independent ethic committee" only. Please provide more details regarding the ethics committee that approved this study. 

The study was approved by the following French ethics committee:

CPP VI CHU G. MONTPIED – Administration centrale – 

58 rue Montalembert

BP 69 – 63003 CLERMONT FERRAND cedex 1, FRANCE

Tél : +33 4 73 75 10 73 – Fax : +33 4 73 75 10 69

mail : cpp-sudest6@chu-clermontferrand.fr

The manuscript and attached files have been revised in accordance with Plos One’s style requirement.

YES. The study was funded by Medtrum France (Boulogne Billancourt, France).

If this statement is not correct, you must amend it as needed.

Amended statement “Medtrum Technologies Inc. Shanghai supported this study. The funders had no role in study design, data collection and analysis. AP (principal investigator) takes full responsibility for the work, including the decision to submit and publish the manuscript. The findings and conclusions in this study are those of the authors and do not necessarily represent the sponsors' views”

I have read the journal's policy and the authors of this manuscript have the following competing interests: C.A. has received writing/consulting honoraria from Medtrum for the present study, declares congress invitations from Eli Lilly, Astrazeneca, and Sanofi; has received speaker honoraria from Diabeloop SA and Eli Lilly; has served on advisory board panels for Eli Lilly. V.M. has received congress invitations from: Novo-Nordisk, Isis, Sadir, Vitalaire, Orkyn, Medtronic, and speaker fees or consultancy for expertise from: Medtrum, ISIS, Novo-Nordis. J.A. has received congress invitations, speaker honoraria, and consultancy fees from Lilly, Sanofi and Medtronic and has served on advisory board panels for Novonordisk, Ypsomed, Medtrum and Glooko. S.C. has received no congress invitations, speaker honoraria, and consultancy fees over the last 24 months. D.G. has received congress invitations, speaker honoraria, and consultancy fees from Eli-Lilly, Novo-Nordisk, Sanofi-Diabète, Abbott, Medtronic, Astra Zeneca. L.C. declares that she has no engagement, fees or honoraria conflicting with her participation to the project. B.C. has received congress invitations, speaker honoraria, and consultancy fees from Medtronic Dexcom and Abbott. P.S.-B. has received conference invitations, honoraria and consultancies from Abbott Diabetes Care, Eli-Lilly, Novo-Nordisk, Roche Diabetes Care and Ypsomed. C.P. has received no congress invitations, speaker honoraria, and consultancy fees over the last 24 months. C.T. has received honoraria for consultancy, speaker for conferences and congress invitations, from Abbott, Janssen, Lilly, Medtronic, Roche Diabetes Care, Sanofi. A.P. has received congress invitations, speaker honoraria, and consultancy fees from Abbott, Diabeloop, Eli Lilly and Company, Insulet, Lifescan, Medtronic, Novo Nordisk, and Sanofi, and has served on advisory board panels for Abbott, Insulet, Medtrum, Novo Nordisk, and Sanofi.

The competing interests section has been updated in line with Plos One requirement.

Ethical and legal restrictions apply to the data as public data-sharing was not planned in the information letter given to and signed by participants at inclusion in the study. Data-sharing would therefore require an amendment of the information given to patients after acceptance by the ethics committee and then participants.

Here is the contact access for the ethics committee that approved the study:

CPP VI CHU G. MONTPIED – Administration centrale – 

58 rue Montalembert

BP 69 – 63003 CLERMONT FERRAND cedex 1, FRANCE

Tél : +33 4 73 75 10 73 – Fax : +33 4 73 75 10 69

mail : cpp-sudest6@chu-clermontferrand.fr

The ethics statement appears only in the Methods section now.

The caption for supporting information (supplemental table one) has been listed at the end of the manuscript as required.

Reviewers' comments:

Reviewer's Responses to Questions

Comments to the Author

1. Is the manuscript technically sound, and do the data support the conclusions?

Reviewer #1: Yes

Reviewer #2: Partly

2. Has the statistical analysis been performed appropriately and rigorously? 

Reviewer #1: Yes

Reviewer #2: Yes

3. Have the authors made all data underlying the findings in their manuscript fully available?

Reviewer #1: Yes

Reviewer #2: No

4. Is the manuscript presented in an intelligible fashion and written in standard English?

Reviewer #1: Yes

Reviewer #2: Yes

5. Review Comments to the Author

Reviewer #1: The article “Performance and patients’ satisfaction with the A7+TouchCare insulin patch pump system: a randomized controlled non-inferiority study” by Amadou and colleagues describes a clinical study with insulin pumps. The study is well-designed and results are presented in a comprehensive manner. Thank you for this nice contribution. I only have some minor comments:

General: I would recommend using the term “glucose management indicator (GMI)” instead of “estimated A1C” as this is state of the art and the methods section references the article by Bergenstal et al. that introduced the GMI

This has been corrected in the text.

Methods: Obviously there was no inclusion criterion that describes a minimum required duration of pump use before study start. According to Table 1 the minimum duration was 0.1 years. This appears too short to be able to differentiate study outcomes from improvements in HbA1c due to change from MDI to CSII.

We understand your concern, but the participants were randomized at baseline, so the effect of the switch from MDI to CSII is supposed to be the same in the two groups. Most patients had been using a pump for 5-7 months. The patients' motivation was to have the opportunity to test an innovative and smaller device. The aim of the study was not to demonstrate superiority of the Medtrum device but to show that the results were similar to Insulet under comparable conditions.

Results and Discussion: It seems like there is a kind of safety issue with the A7+TouchCare pumps, because 7 patients dropped out due to intolerance or device dysfunction and in general, there were more adverse events with this pump. I think this is should be paid more attention in the discussion of the article.

There are indeed more AEs and dysfunctions reported in the A7+ Touchcare group. It appeared after discussion with the centres that patients who had already been using the Omnipod pump for several months did not report all events because they were used to their occurrence. On the other hand, in the A7+ Touchcare group, patients who received minimal and rapid training did not always have a perfect mastery of the devices and probably had a more systematic notification of events. Furthermore, it should be remembered that the study was carried out in the context of the covid epidemic, with support from the health care teams that was somewhat different from a normal context, which may have increased the anxiety of the patients in the A7+ Touchcare group when events occurred.

Author contributions: All investigators all listed as authors, but most of them did not make any further contributions to the manuscript. This is not conform with the ICMJE authorship criteria; e.g. critically revising the manuscript is another required contribution.

The manuscript has been sent for review and validation to all co-investigators. The author contribution statement has now been updated.

Reviewer #2: Here are some comments:

Statistical methods:

“In addition, univariate analyses were performed on age in class (according to

the median value), gender, diabetes mellitus type, center, and duration of diabetes. Variables with a p-value less than 0.05 were included in the multivariate model using a stepwise strategy.” My comment here is that conclusions on these variables should be drawn from a full model i.e. a multiple model including all variables. The stepwise strategy is not needed and if any adjustment for the primary analysis is done, why not adjust for all?

Indeed, only one variable was significant at the 5% level, so we removed the stepwise strategy in the article since it was considered but not used.

Table 1. Why include statistical tests. Since this is a randomized trial this is not needed. Read about “the table 1 fallacy” and point 15 in the consort checklist http://www.consort-statement.org/checklists/view/32--consort-2010/510-baseline-data.

You are right, the p-values should not be displayed in table 1. We removed it from the table.

How was the p-value for non-inferiority calculated (table 2)? Probably through the lsmeans statement, include this in the methods.

We have added this precision in the text.

Table S1: Please indicate which test that is used and double check the p-values, is the result for “Redness” correct?

Thank you for your attention. The value for redness was wrong (but still not significant) and has been modified.

Why is the “other adverse events” not discussed and is the conclusion “Overall, the A7+TouchCare may be a safe and valuable alternative to Omnipod..” correct?

Only the main adverse events usually described in the use of pumps and sensors were retained, the other AEs being minor. In addition, it should be noted that there is a potential bias because the patients in the Omnipod pump group had all been using the system for several months prior to inclusion and were therefore already sufficiently tolerant. This had the consequence of reducing or even not reporting tolerance problems that patients in this group are used to. We agree to change the conclusion sentence to “Overall, the A7+TouchCare may be a valuable alternative to Omnipod or other insulin patch pumps in some patients with diabetes.”

Minor: The following sentence is repeated: “The inclusion of patients already using a pump was done to facilitate recruitment into the study. The inclusion of patients already using a pump was done to facilitate recruitment into the study.”

This has now been corrected in the text.

6. PLOS authors have the option to publish the peer review history of their article (what does this mean?). If published, this will include your full peer review and any attached files.

Do you want your identity to be public for this peer review? For information about this choice, including consent withdrawal, please see our Privacy Policy.

Reviewer #1: No

Reviewer #2: No

---

## [Decision Letter · Decision Letter 1]

16 Jun 2023

PONE-D-22-17489R1Performance and patients’satisfaction with the A7+TouchCare insulin patch pump system: a randomized controlled non-inferiority studyPLOS ONE

Dear Dr. Amadou,

Thank you for submitting your manuscript to PLOS ONE. After careful consideration, we feel that it has merit but does not fully meet PLOS ONE’s publication criteria as it currently stands. Therefore, we invite you to submit a revised version of the manuscript that addresses the points raised during the review process.

We look forward to receiving your revised manuscript.

Kind regards,

Aleksandra Klisic

Academic Editor

PLOS ONE

Journal Requirements:

Reviewers' comments:

Reviewer's Responses to Questions

**Comments to the Author**

1. If the authors have adequately addressed your comments raised in a previous round of review and you feel that this manuscript is now acceptable for publication, you may indicate that here to bypass the “Comments to the Author” section, enter your conflict of interest statement in the “Confidential to Editor” section, and submit your "Accept" recommendation.

Reviewer #2: All comments have been addressed

Reviewer #3: All comments have been addressed

2. Is the manuscript technically sound, and do the data support the conclusions?

Reviewer #2: (No Response)

Reviewer #3: Yes

3. Has the statistical analysis been performed appropriately and rigorously? 

Reviewer #2: (No Response)

Reviewer #3: Yes

4. Have the authors made all data underlying the findings in their manuscript fully available?

Reviewer #2: (No Response)

Reviewer #3: Yes

5. Is the manuscript presented in an intelligible fashion and written in standard English?

Reviewer #2: (No Response)

Reviewer #3: Yes

6. Review Comments to the Author

Reviewer #2: (No Response)

Reviewer #3: PONE-D-22-17489R1

This is an interesting and well written study, as we lack RCTs regarding the achievements of insulin patch pumps. The current report adresses both performance and satisfaction with a new insulin patch pump in an appopriate, randomized design.

My main concern is that the abstract does not mention about the flow chart and premature study withdrawals, which were more frequent in the Touchcare group, leading to a desequilibrium between the two groups in the per protocol analysis.

Regarding satisfaction, the sentence on page 12 about the preference score is unclear, as the insulin management system, opposed to the patch insulin delivery system and the PDM, have not been clearly defined in the Materials section.

Minor comment: in the Method section, please state whether the fast insulin analog URLI was allowed or not.

7. PLOS authors have the option to publish the peer review history of their article (what does this mean?). If published, this will include your full peer review and any attached files.

Reviewer #2: No

Reviewer #3: No

---

## [Author Response · Author response to Decision Letter 1]

6 Jul 2023

Dear Editor,

The manuscript has been revised. 

You will find answers to each comment below (italics and bold).

Major and minor corrections (responding to comments) have been highlighted in yellow in the revised manuscript (“Revised Manuscript with Track Changes”). We also provide a final clean version (“'Manuscript”).

We are confident that the quality of the manuscript is now improved, and we hope our revised version will be considered suitable for publication. 

The original comments of reviewers are written in blue and our response in black just below.

This is an interesting and well written study, as we lack RCTs regarding the achievements of insulin patch pumps. The current report adresses both performance and satisfaction with a new insulin patch pump in an appopriate, randomized design.

My main concern is that the abstract does not mention about the flow chart and premature study withdrawals, which were more frequent in the Touchcare group, leading to a desequilibrium between the two groups in the per protocol analysis.

We have modified the abstract and added the number of premature withdrawals in each group.

Regarding satisfaction, the sentence on page 12 about the preference score is unclear, as the insulin management system, opposed to the patch insulin delivery system and the PDM, have not been clearly defined in the Materials section.

We agree that the sentence is not clear. The insulin management system designs the patch insulin delivery system and the PDM as a whole. We propose to modify the sentence by writing: “The preference score was 3.7±1.5 for the patch insulin delivery system, 1.9±1.3 for the PDM and 3.2±1.4 for the insulin management system (designing the patch insulin delivery and PDM as a whole).

Minor comment: in the Method section, please state whether the fast insulin analog URLI was allowed or not.

The fast insulin analog URLI was not allowed in this study as it was not available in France at the moment of the study protocol development.

Dr Coralie Amadou, for all authors.

---

## [Editor Report · Decision Letter 2]

25 Jul 2023

Performance and patients’satisfaction with the A7+TouchCare insulin patch pump system: a randomized controlled non-inferiority study

PONE-D-22-17489R2

Dear Dr. Amadou,

We’re pleased to inform you that your manuscript has been judged scientifically suitable for publication and will be formally accepted for publication once it meets all outstanding technical requirements.

Kind regards,

Aleksandra Klisic

Academic Editor

PLOS ONE

Additional Editor Comments (optional):

Dear Author,

The manuscript PONE-D-22-17489R2

"Performance and patients’ satisfaction with the A7+TouchCare insulin patch pump system: a randomized controlled non-inferiority study" by Dr Coralie Amadou is accepted for publication in PLOS ONE.

Sincerely,

Aleksandra Klisic

---

## [Editor Report · Acceptance letter]

14 Aug 2023

PONE-D-22-17489R2 

Performance and patients’satisfaction with the A7+TouchCare insulin patch pump system: a randomized controlled non-inferiority study 

Dear Dr. Amadou:

I'm pleased to inform you that your manuscript has been deemed suitable for publication in PLOS ONE. Congratulations! Your manuscript is now with our production department. 

Kind regards, 

on behalf of

Dr. Aleksandra Klisic 

Academic Editor

PLOS ONE